# Intelligent Sight and Sound:
# A Chronic Cancer Pain Dataset

**Catherine Ordun** [1,2]**, Alexandra N. Cha** [1]**, Edward Raff** [1,2]**,**
**Byron Gaskin** [1]**, Alex Hanson** [1]**, Mason Rule** [3]**, Sanjay Purushotham** [2]**, James L. Gulley** [3]
[1] Booz Allen Hamilton
[2]University of Maryland, Baltimore County
[3]Center for Cancer Research, National Cancer Institute, National Institutes of Health

## Abstract

Cancer patients experience high rates of chronic pain throughout the treatment process. Assessing pain for this patient population is a vital component of psychological and functional well-being, as it can cause a rapid deterioration of quality of life. Existing work in facial pain detection often have deficiencies in labeling or methodology that prevent them from being clinically relevant. This paper introduces the first chronic cancer pain dataset, collected as part of the Intelligent Sight and Sound (ISS) clinical trial, guided by clinicians to help ensure that model findings yield clinically relevant results. The data collected to date consists of 29 patients, 509 smartphone videos, 189,999 frames, and self-reported affective and activity pain scores adopted from the Brief Pain Inventory (BPI). Using static images and multi-modal data to predict self-reported pain levels, early models show significant gaps between current methods available to predict pain today, with room for improvement. Due to the especially sensitive nature of the inherent Personally Identifiable Information (PII) of facial images, the dataset will be released under the guidance and control of the National Institutes of Health (NIH).

## 1   Introduction

The prevalence of chronic pain in cancer patients is high, with an estimated prevalence of 59% in those undergoing anticancer treatment, 64% of whom have advanced stage disease and 33% who continue to experience pain following completion of curative treatment [1]. Despite advances in pain management, prompt assessment and management of cancer pain remains a challenge and a large proportion of patients continue to experience moderate to severe pain.

Sub-optimal pain management can block patient recovery and improvement, making the already difficult cancer experience, worse, for both patient and family [2, 3]. Manual clinical assessment requires accounting for a landscape of complex emotions and beliefs that clinicians must regularly take into account when assessing cancer patient pain - physical, psychological, social, and spiritual elements combined with severe distress for future outlook [2, 4]. For example, patients undergoing chemotherapy are more likely to believe that "good patients" do not complain about pain which they believe can be distracting to clinicians and become non-communicative [2]. Further, few patients are actually screened for pain at each clinical visit [3], and pain is under-reported in patient populations such as nursing home patients [3]. Due to the variety of complex conditions affecting cancer pain, experts recommend repeated, regular pain assessment, which can be difficult and impractical for manual assessment by clinicians.

Currently, no facial pain datasets exists for chronic cancer pain and little research overall has been conducted into machine learning for the identification and evaluation of chronic pain. For example, in

Submitted to the 35th Conference on Neural Information Processing Systems (NeurIPS 2021) Track on Datasets and Benchmarks. Do not distribute.

a review by [4], only seven out of 52 machine learning papers evaluated pain in a non-acute context such as chronic fatigue, fibromyalgia, chronic pancreatitis, migraines, and genetic pain. Existing facial pain research focuses on acute, musculoskeletal pain such as chronic lower back pain [5] and shoulder pain [6, 7] or simulated pain induced by heat or electrical stimuli [8, 9] where painful expressions are obvious through grimaces and eye raises. Such datasets are manually labeled by trained observers with Facial Action Coding Units [10], making the labeling procedure prohibitively expensive and impractical for clinical use. Further, external pain labels may be biased towards an outside observer's impression of a patient's pain, not the patient themselves. Research also shows that typical pain facial expressions that correlate with physical pain are less frequently observed among chronic pain cancer patients who exhibit subdued and placid expressions [11].

Given the limitations of existing facial expression pain data, the U.S. National Institutes of Health (NIH) National Cancer Institute (NCI) initiated *"A Feasibility Study Investigating the Use of Machine Learning to Analyze Facial Imaging, Voice and Spoken Language for the Capture and Classification of Cancer Pain"* [12], or "Intelligent Sight and Sound" (ISS). Details of the protocol are available publicly at `https://clinicaltrials.gov/ct2/show/NCT04442425`. This is an observational, non-interventional clinical study that aims to address the following problem statement [12, 13]: *"To determine if a new observational based pain prediction algorithm can be produced that is accurate to standard, patient-reported pain measures and is generalizable for a diverse set of individuals, across sexes and skin types."* The study has two objectives: 1) investigate facial image data, and 2) analyze text and audio, as modalities for predicting self-reported chronic cancer pain.

The study is ongoing and aims to recruit 112 patients. We report the initial dataset, which is less than a quarter of the final data consisting of 29 patients. Data include multimodal extracts from video submitted in a spontaneous, home setting, and in a few cases of in-clinic capture at the NIH. It includes visual spectrum (RGB) video frames, facial images resulting from face detection models, facial landmarks from Active Appearance Models (AAMs) [14, 15], audio files, Mel spectrograms, audio features, and self-reported pain scores adopted from the Brief Pain Inventory (BPI) [16–18].We will present details of the study design, data distribution, and storage procedures to ensure patient privacy. We also provide initial baseline results for pain classification using simple, traditional, machine learning models and neural networks.

## 2 Related Works

Automatically detecting pain from facial expressions has been extensively published following methods of facial emotion recognition (FER). The majority of these works have focused on acute or musculoskeletal physical pain [4, 19–27]. Primary pain datasets based upon facial imaging include UNBC-McMaster Shoulder Pain Expression Archive [6, 7], the Biopotential and Video Heat Pain (BioVid) Database using controlled, simulated heat to induce pain [8], Multimodal Intensity Pain (MIntPAIN) database using pain resulting from electrical stimulation [9], the Experimentally Induced Thermal and Electrical (X-ITE) Pain Database [28, 29], and the EmoPain for chronic, musculoskeletal pain [30]. These datasets traditionally contain video sequences since video enables continuous clinical monitoring of pain response [18]. These datasets also contain extensive offline annotations of pain ratings by external observers, and sometimes include additional modalities such as thermal and depth data. Additional video facial expression pain datasets exist that focus on different patient populations, but primarily focus on physical pain. These include multimodal behavioral and physiological data for neonatal pain [31, 32] and the University of Regina (UofR) Pain in Severe Dementia dataset [33, 34]. A summary of the pain datasets is provided in Table 1.

Table 1: **Related Pain Datasets.**

| Dataset | Stimulus | Subjects | Frames | Sequences | Seq. Duration | Modality |
|---|---|---|---|---|---|---|
| UNBC-McMaster [6, 7] | Shoulder pain | 25 | 48,398 | 200 | 10 - 30 sec., per | Unimodal |
| BioVid [8] | Heat stimulus | 90 | 8700 | 87 | 5.5 sec. | Multimoda |
| MIntPAIN [9] | Electronic stimulus | 20 | 187,939 | 9366 | 1 - 10 sec. | Multimodal |
| EmoPain [30] | Chronic lower back pain | 22 | 44,820 | 35 | 3 sec. | Multimodal |
| Neonatal Pain, USF [32, 35, 36] | Heel lancing | 31 | 3026 | 200 | 9 sec. | Multimodal |
| UofR [33] | Physical, painful movements | 102 | 162,629 | 95 | Unknown | Multimodal |
| X-ITE [28, 29, 37] | Heat and electronic stimuli | 134 | 26,454 | N/A | 7 sec. | Multimodal |
| **ISS (Dec. 2020 - Jul. 2021)** | **Chronic cancer pain** | **29** | **189,999** | **509** | **3.52 - 135.59** | **Multimodal** |

# 3 ISS Dataset

The ISS protocol is a single site study with a goal of enrolling a total of 112 patients (90 adult and 22 pediatric) who are actively receiving treatment for advanced malignancies and/or tumors at the NIH Clinical Center or treated with standard of care in the community. The study is overseen by the NIH Institutional Review Board (IRB), and the protocol was also reviewed by NCI's Center for Cancer Research (CCR) Scientific Review Committee. New patient enrollment was paused during the Covid-19 pandemic due to initially unknown risks, but has resumed with vaccine availability and clinician guidance.

## 3.1 Sample and Study Design

To obtain as representative a sample as possible within the constraints of a feasibility study with an overall small sample size, the sample consists of twelve cohort groups of seven patients each. Patients represent a breadth of age, sex, skin tone (as a proxy for ethnicity), and pain experience. The current ISS dataset consists of 29 adult patients ages 18 years and over who have consented to participate in the study; no pediatric patients (ages 12-17) have been enrolled yet.

The goal is to evenly split the sample by i) sex (Male or Female), ii) Fitzpatrick Skin Type [38], a self-reported, visual method of skin tone classification, where patients are asked to type themselves into one of two groups: "light" skin tones in types I-III or "dark" skin tones in types IV-VI, and iii) a self-reported "worst" pain score reported on a 0 – 10 Numerical Rating Scale (NRS) [39]. The

Table 2: **ISS: Twelve Patient Cohorts.**

| Number | Pain Target | Skin Types | Sex | Pain Class | Goal | Current |
|---|---|---|---|---|---|---|
| 1A | 0 | I - III | Male | None | 7 | 7 |
| 1B | 0 | I - III | Female | None | 7 | 2 |
| 1C | 0 | IV - VI | Male | None | 7 | 4 |
| 1D | 0 | IV - VI | Female | None | 7 | 0 |
| 2A | 1-3 | I - III | Male | Low | 7 | 1 |
| 2B | 1-3 | I - III | Female | Low | 7 | 1 |
| 2C | 1-3 | IV - VI | Male | Low | 7 | 3 |
| 2D | 1-3 | IV - VI | Female | Low | 7 | 0 |
| 3A | 4-6 | I - III | Male | Moderate | 7 | 2 |
| 3B | 4-6 | I - III | Female | Moderate | 7 | 2 |
| 3C | 4-6 | IV - VI | Male | Moderate | 7 | 0 |
| 3D | 4-6 | IV - VI | Female | Moderate | 7 | 0 |
| 4A | 7-10 | I - III | Male | Severe | 7 | 1 |
| 4B | 7-10 | I - III | Female | Severe | 7 | 2 |
| 4C | 7-10 | IV - VI | Male | Severe | 7 | 2 |
| 4D | 7-10 | IV - VI | Female | Severe | 7 | 3 |
| | | | | | **112** | **30** |

self-reported pain score is referred to as the "Pain Target" and are grouped into levels 0, 1-3, 4-6, and 7-10. It represents the worst pain the patient has experienced in the past thirty days prior to the start of the study. It is fixed throughout the patient's enrollment and does not change. As a result, there is no variance for the "Pain Target" score. The "Pain Target" is the classification target which is later used for our baseline tasks.

The twelve different cohorts are shown in Table 2, along with the goal of seven patients to be enrolled per cohort, and the current distribution of patients enrolled. Note that data from one patient (0009) in Cohort 2B was unusable. As a result our analysis reports across 29 patients. Clinical inclusion criteria include individuals with a diagnosis of a cancer or tumor who are under active treatment for this condition at NIH/NCI. Patients must also have access to a smart phone or computer with camera, microphone, and internet access. Several clinical exclusion criteria apply. Excluded are patients with active central nervous system (CNS) metastases, with the exception

Table 3: **ISS Study Design Overview.** * Note that due to the global Covid-19 pandemic, the majority of patient videos were submitted in the remote setting.

| | |
|---|---|
| Study Duration | 3 months |
| # Remote Submission | 3 / week; Max. 1 /day |
| # In-Clinic Submissions | 1 - 4 /week* |
| Survey Tool | Smartphone (iPhone or Android) or computer (camera/mic.) |
| Time / Submission | Average total time: 3 min |
| Time / Question | Q 1-9: 1 min, Q10: 15 sec, Q11: 15 sec. - 3 min. |
| Self-Reported Pain | Q 1-9: Questions with Likert-scale responses |
| Voice/Video: Prompt | Q10: Read and record one of 3 randomized nursery rhymes. |
| Voice/Video: Narrative | Q11: Record respond to "Describe how you feel right now." |
| Compensation: | Min. 3 / week, they earn $15. |

of those who have completed curative intent radiotherapy or surgery and have been asymptomatic for three months prior to consent, patients with Parkinson's disease, and any psychiatric condition that would prohibit the understanding or rendering of informed consent. Additional exclusion criteria those who are non-English speaking or have known current alcohol or drug abuse. Each patient is enrolled for a three-month period and are financially incentivized to complete three check-ins per week remotely and up to four in-clinic check-ins. The study design is summarized in Table 3. Patients engage using an electronic questionnaire and through video recording using a custom developed mobile or web application, using an Android, iPhone, or computer with camera and microphone.

## 3.2   Patient Protocol

Figure 1 provides a series of screenshots showing the patient at-home or in-clinic check-in using the ISS application. For each approximately 3-minute check-in, patients respond to a nine element questionnaire based on the Brief Pain Inventory (BPI, licensed from MD Anderson) [16–18] and two prompts to record videos of themselves.

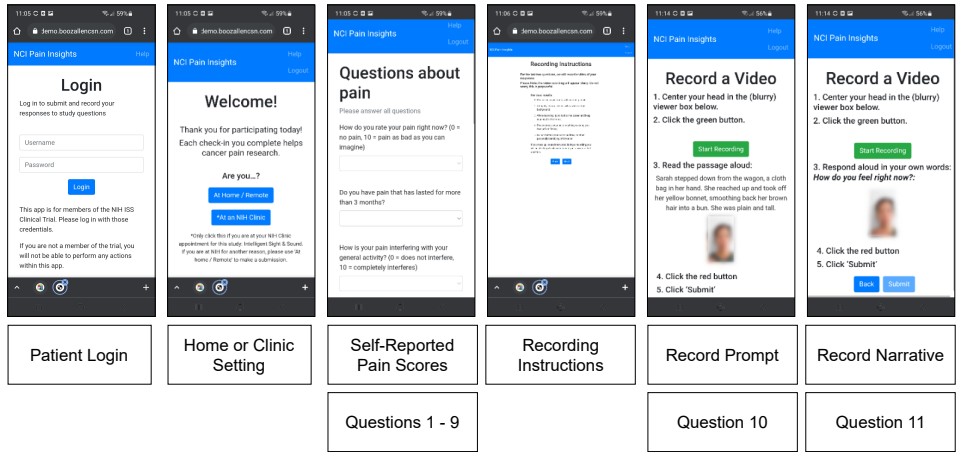

Figure 1: **Submitting a Video through the ISS Mobile Application.**

### 3.2.1   Questions 1 - 9: Self-Reported Pain Scores

In addition to the self-reported "Pain Target" which was assigned to each patient upon enrollment shown in Table 2, there are nine additional self-reported pain scores. We capture these self-reported pain scores based on the cancer pain literature which indicates that cancer patients experience complex emotions and beliefs that can influence their perception of pain and as a result, clinical treatment [2, 3]. These nine pain scores are submitted at the time of video submission and change at each submission. They are distinct and unrelated to the "Pain Target" which is used for the baseline classification tasks. There is no formula that relates the nine self-reported scores among themselves or to the "Pain Target". In the below, Question 1 captures current pain intensity scored on an 11-point Likert Scale (0 No Pain - 10 Worst Possible Pain) followed by Question 2 which, when answered affirmatively, indicates the presence of chronic pain. Questions 3-9 utilize an 11-point Likert Scale (0 Does not Interfere - 10 Completely Interferes) to measure the interference of pain in an individual's activity (3, 5, 6) and an individual's affect or mood (4, 7, 8, 9).

1. How do you rate your pain right now? (0 No Pain – 10 Worst Possible Pain on Likert-scale).
2. Do you have pain, related to your cancer, that has lasted for more than 3 months? (Yes/No)
3. How is your pain interfering with your General Activity?
4. How is your pain interfering with your Mood?
5. How is your pain interfering with your Walking Ability?
6. How is your pain interfering with your Normal Work (both work outside the home and housework)?
7. How is your pain interfering with your Relationships with other people?
8. How is your pain interfering with your Sleep?
9. How is your pain interfering with your Enjoyment of life?

### 3.2.2 Questions 10 and 11: Prompt and Narrative

Following the questionnaire, Question 10 is a prompt to record a video where the patient reads a 10-15 second passage of text at a grade 3 reading level selected at random from three different passages. The use of this sort of prompt is common practice in mood induction or conditioning trials where a neutral, non-emotion inducing prompt is used as a control versus a potentially, emotionally charged response related to the experimental condition [40–42]. The neutral passage options are:

- "Sarah stepped down from the wagon, a cloth bag in her hand. She reached up and took off her yellow bonnet, smoothing back her brown hair into a bun. She was plain and tall." From Sarah Plain and Tall by Patricia MacLachlan [43]
- "And then the dog came running around the corner. He was a big dog. And ugly. And he looked like he was having a real good time. His tongue was hanging out and he was wagging his tail." From Because of Winn Dixie by Kate DiCamillo [44]
- "You have brains in your head. You have feet in your shoes. You can steer yourself any direction you choose. You're on your own. And you know what you know. And YOU are the one who'll decide where to go." From Oh the Places You Will Go by Dr. Seuss [45]

Finally, in Question 11, the patient records a video responding to the prompt *"Please describe how you feel right now."* Narratives include discussion of medical conditions, mood, daily activities, current beliefs and attitudes about their pain. The allowable video length can range from 15 seconds to 3 minutes, with recording instructions shown prior to each video prompt. For "at-home" check-ins, patients are instructed to complete the submission alone, in a quiet and brightly lit room, preferably with a white wall or background. In addition, patients are asked not to reveal personal information such as their name or address. In Figure 1, the application screens for Questions 10 and 11 include a live video image to help the patient keep their face centered in the frame, but the application blurs the video. The blur effect is to prevent the patient from manipulating their facial expression and minimize self-conscious alteration of their appearance, allowing them to focus on their responses.

### 3.3 Data Description

A high level summary of the ISS dataset is provided in Table 4. The ISS dataset is comprised of 29 patients submitting videos in a spontaneous, non-posed, home setting through a smartphone or computer. Patients are adults over the age of 18 y.o. and consist of the following demographics: 20 Male, 9 Female, 17 Skin Type I-III, 12 Skin Type IV - VI. All patients were enrolled between December 2020 and July 2021. There are 189,999 total video frames. After facial detection, we extracted 173,011 facial images. After landmark detection on the facial images, the dataset was reduced by 2.86% to 168,063 facial images with landmarks, since landmarks could not be detected for some faces. We show the ratio of data imbalance across four pain levels using the total frames in Table 4 where the "None" label is the majority class. The dataset also contains self-reported pain scores from Questions 1 - 9, described in detail in the Study Design section, along with sex and skin type labels assigned upon enrollment. Additional descriptive analysis is provided in Supplementary Materials.

Table 4: **ISS Data Summary.**

| ISS Data Summary | | | | Ratios of Total Frames by Pain Levels | | | | | | | |
|---|---|---|---|---|---|---|---|---|---|---|---|
| Total Patients | 29 | 20 M, 9 F, 17 Skin Type I-III, 12 Skin Type IV-VI | | 4 Pain Levels | Frames | Ratio | No. Patients | 2 Pain Levels | Frames | Ratio | No. Patients |
| Total Videos | 509 | Avg. Videos per Patient | 17.55 | None | 100984 | 1.00 | 13 | No Pain | 100984 | 1.00 | 13 |
| Total Frames | 189,999 | Avg. Frames per Patient | 6551 | Low | 11784 | 8.57 | 4 | Pain | 89015 | 1.13 | 16 |
| Total Duration | 316 min. | Avg. Duration per Patient | 655 sec. | Mod. | 25999 | 3.88 | 4 | | | | |
| Avg. Duration per Video | 37.32 sec. | Range of Duration per Video | 3.52 - 135.79 sec. | Severe | 51232 | 1.97 | 8 | | | | |

A notional depiction of ISS data types is shown in Figure 3 to provide context for the data types. Due to the sensitivity of Personal Identifiable Information (PII) in the clinical study protocol, we are unable to display actual facial images from the dataset at this time.

### 3.3.1 Data Extraction

We use the patient narrative (Question 11) video files (.mp4) and extract frames at 10 frames-per-second. We decide to use the narrative versus the prompt since it may contain greater signals of pain and emotion, compared to the neutral baseline recording. An audio .wav file of the patient narrative is simultaneously extracted using the ffmpeg library. We use the PyTorch FaceNet library that

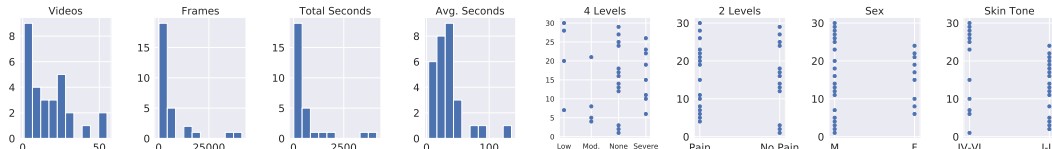

Figure 2: **Distribution of ISS Data.** Histograms for the total videos, frames, seconds, and average seconds per video, for the ISS dataset are in the four left-most plots. The four plots on the right illustrate the distribution of patients (y axis) by the four pain levels, when combined into two pain levels, by sex, and by skin type.

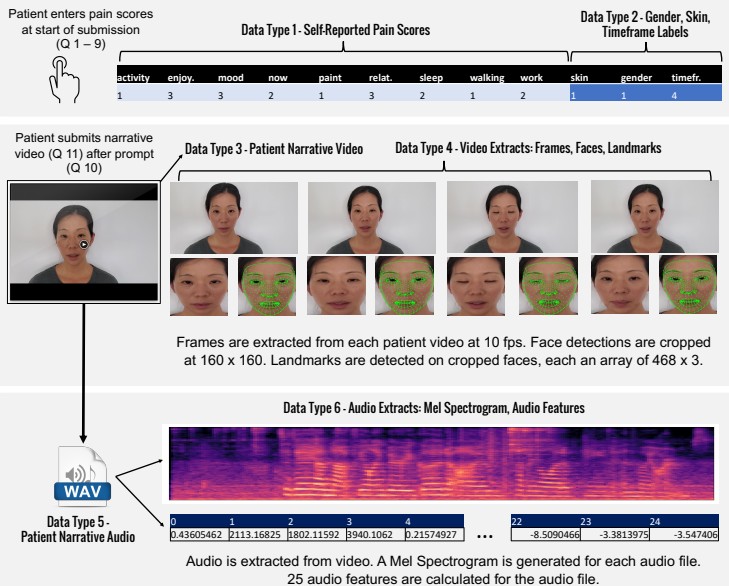

Figure 3: **ISS Data Types.** Facial images shown are *not* actual patients from the ISS dataset due to privacy restrictions. The ISS Dataset currently consists of six types of data: 1) Nine Self-Reported Pain Scores, 2) Labels for Sex, Skin Type, and Timeframe, 3) Patient Narrative Video, 4) Video Extracts: Frames, Faces, Landmarks, and 5) Patient Audio, and 6) Audio Extracts: Mel Spectrogram, Audio Features.

implements a fast and CUDA-enabled version of the Multi-task Cascaded Convolutional Networks (MTCNN) algorithm [46] using an InceptionResnetV1 model pre-trained on VGGFace2 for face detection and cropping faces from frames. All patient faces were recorded in a frontally aligned position so no realignment was implemented. Similar to [26], we extract features using AAMs. Specifically, we use the Google MediaPipe [47] Face Mesh AAM model based on 3D Morphable Models [15] to detect facial landmarks where each face returns an array of 468 points for three coordinates. From the audio .wav file, we use the Librosa [48] library to generate a Mel Spectrogram (n_fft=2048, hop_length=512, n_mels=128), and apply signal processing to capture audio features about the .wav file to include Mel-frequency cepstral coefficients (MFCCs), chromogram, spectral centroid, spectral bandwidth, roll-off frequency, and zero crossing rate, leading to 25 audio features. We further break up the original video into 4-second chunks leading to 40 frames per video chunk, extracting its respective .wav file, spectrogram, and audio features.

### 3.3.2 No External Labels

In contrast to existing acute pain datasets [7–9], the ISS dataset lacks external offline labeling traditionally completed using the Facial Action Coding System (FACS) [10]. Per the ISS problem statement, the goal is to predict patient (self)-reported pain, as opposed to observations made by non-patients via offline pain coders. There are three reasons for not externally encoding ISS video frames using FACS. First, researchers agree that FACS is expensive due to the need for a trained coder to annotate each video frame, making the process time-consuming and clinically infeasible [18, 49]. Second, ethicists and psychologists argue that there is limited evidence that facial expressions

are reliably and specifically mapped to emotion production [50, 51]. Emotion production is not necessarily tied to a single set of facial expressions, but relies on the context of the situation and human culture [50]. Third, cancer patients with chronic pain may not display the typical set of facial action units (AUs) commonly associated with acute pain. For example [11] collected video data from 43 outpatient lung cancer patients obtained in a spontaneous home setting [11]. They found that the cancer patients were more subdued in expression, and displayed fewer AUs such as grimaces or clenched teeth, commonly found in facial pain images. As a result, AU labels associated with pain such as brow lowering (AU4), orbital tightening (AU6, AU7), levator contraction (AU9, AU10), and eye closure (AU43) may not be applicable to chronic cancer facial pain detection [52]. However, when the ISS dataset is released, there are no prohibitions on researchers attempting to annotate using FACS.

### 3.4 Data Storage and Access

A secure cloud-based environment receives mobile and web-based submissions of patients' video, audio, and survey (nine self-reported pain scores) data. No PII such as names or date of birth is stored, with the exception of face, voice, and sex information. The environment is AWS GovCloud FedRAMP Moderate, with Federal Information Security Management Act (FISMA) moderate Authority-to-Operate (ATO) credentials.

The ISS dataset consists of cancer patients discussing their medical conditions. The very nature of the images and videos make the data Protected Health Information (PHI) due to the NIH/NCI not being classified as a "covered entity". Extreme care must be exercised to ensure patient privacy and rights are not violated. As a result, we plan to ensure proper patient protections by placing the collected data in restricted access repositories under the stewardship of the NIH. Members of the scientific community will be able to request access to the data and code which may be granted on a per-case basis. This requirement is necessary to ensure legal requirements are met, avoid public spillage of PII data, and ensure patient trust that their data is used within the scope of the intended scientific use. In return researchers receive access to a dataset with numerous modalities and potential clinical relevance of results.

## 4 Baselines

We conduct seven baseline experiments for a classification task to predict each patient's self-reported "Pain Target" level assigned at the start of their enrollment shown in Table 2. These levels are fixed upon enrollment for cohort assignment and remain unchanged throughout the study. As a result, one patient represents a single pain level throughout the study. All experiments are static models, which return predictions on a frame-by-frame level. Given how we are in the initial phase of the ISS study, we train models using facial images, landmarks, and the additional nine self-reported pain scores for emotion and activity. However, we do provide baseline results on 4-second chunks of audio via spectrograms and audio features. These are meant to be representative of common approaches to similar work, and establish the careful curation results in a task more difficult than prior literature with simpler labeling or collection. More details on all results are in the Supplementary Materials.

**Training Details** All experiments are trained using 10-fold cross validation where three test patients are withheld in the test set for nine splits and two patients set aside for the tenth split. There is no overlap between training and test sets for each split. Please refer to the Supplementary Materials Appendix Section F.1. Table 10 that shows the "10-fold-CV details - Test Patients per Split." For neural networks in Experiments 1 and 3 - 7, we use a batch size of 16, Adam optimizer with 1e-4 learning rate, and cross entropy as the loss function, training for 10 epochs, for all experiments. The batch size of 16 was selected empirically based on cross validation accuracy, after running several experiments varying batch size from 4, 8, 16, 32, and 64. We selected Adam optimizer since it has been used in recent facial pain detection studies such as [24, 53]. We fine-tune ResNet50 as the convolutional neural network (CNN) backbone for all multimodal experiments, which is pretrained on ImageNet. We use PyTorch for model training and train on four NVIDIA Tesla T4 GPUs. Experiments 2 and 3 are trained using the Scikit-Learn library for the Random Forest Classifier, using 100 estimators, gini criterion, min_samples_split=2, and min_samples_leaf=1.

**Experiment 1: Pain Prediction using Static Face Images** The first set of experiments only uses static, facial images. We fine-tune ResNet-50 [54] pretrained on ImageNet [55] to predict four and two levels of pain. Four levels are "None" (Self-Reported Pain Level 0), "Low" (1-3), "Moderate"

(4-6), or "Severe" (7-10), and two levels combine "Low", "Moderate", and "Severe" pain levels into a single "Pain" class. Training binary classifiers for"No Pain"/"Pain" prediction is similar to many existing facial pain detection works [7, 22]. We found that the binary classifier leads to better test patient accuracy scores, and continued Experiments 2 through 7 using only two pain levels.

**Experiments 2, 3: Pain Prediction using Static Landmarks or Pain** In these experiments, we use only one modality to train two separate models and use traditional machine learning models, specifically the Random Forest algorithm [56]. Experiment 2 uses the landmark arrays detected for each facial image and Experiment 3 uses the nine self-reported pain scores explained in Section 3.2.1 that represent how pain interferes with the patient's emotions and activity, plus labels for sex, skin type, and timeframe. The timeframe label is categorical and is extracted from the video submission timestamp representing what time of day (early AM, late PM, etc.) the video was submitted. For both Experiments 2 and 3, we train a Random Forest Classifier. Note, that the target "Pain Target" is not in the set of the nine self-reported pain scores, which are distinct and separate.

**Experiments 4 - 6: Pain Prediction using Static Multimodal Data** We train three multimodal networks using an early, joint fusion strategy as proposed by [57]. For Experiment 4 ("Fusion 1"), we concatenate the fully connected outputs of ResNet50 with raw landmarks. The feature vector is then inputted to a feedforward neural network for binary pain prediction. Experiment 5 ("Fusion 2") concatenates the fully connected outputs of ResNet50 with raw landmarks, in addition to the nine pain scores, skin, sex, and timeframe labels. Similarly, the feature vector is inputted to the same feedforward network architecture for binary pain prediction. Experiment 6 ("Fusion 3") concatenates three vectors: the feature map from `layer-4-conv2D-1`, the landmark features outputted from a landmark-specific feedforward network, and the nine pain scores, sex, skin, and timeframe features outputted from a pain-specific feedforward network. The resulting feature vector is inputted to a CNN for binary pain prediction.

**Experiment 7: Audio Models** Experiment 7 is a binary pain prediction model that uses the Mel spectrogram image and 25 audio features from 4-second chunks of audio extracted from each patient video. A feature vector resulting from the concatenation of the spectrogram feature map from `layer-4-conv2D-1` and audio features learned by a feedforward network, are inputted to the same CNN architecture as used in Experiment 6. Diagrams for all experimental architectures are provided in the Supplementary Materials.

# 5 Results

**Accuracy Calculation** The accuracy of each model is evaluated for each test patient using the tenth model checkpoint. Using the checkpoint, we evaluate each test patient individually. We only evaluate test patients using their respective, assigned split per 10-fold cross-validation (See Supplementary Materials Section F.1. Table 10 "10-fold-CV details - Test Patients per Split" for details). For example, test patients 0002, 0029, and 0021 are only evaluated using the trained model from Split 1, not Split 2 which would have included these three patients in its training set. We evaluate each test patient using a batch size of 1, predicting the target pain score for each patient image. We then calculate accuracy for the test patient in question as simply $accuracy\_score(y\_true, y\_pred)$ where $y\_true$ is the set of true "Pain Target" labels and $y\_pred$ is the set of predicted "Pain Target" labels.

As a result, in Table 5, we show the mean accuracy computed for each "Pain Target" level across all test patients ("No Pain" or "Pain" for two levels, and "None", "Low", "Moderate", or "Severe" for four levels of pain). For example, in Experiment 1 "ResNet50-4-static", the accuracy scores for all patients with ground truth pain labels of "None", were averaged together to calculate the result of 0.583. In Figures 4 and 5, the bars are color-coded by the ground truth "Pain Target" level for each patient. The y-axis is the accuracy predicted for the patient. For example, upon zooming into Figure 4a, Patient 0029's (8 marks from the right of the x-axis) ground truth "Pain Target" level is "No Pain". However, the Experiment 1 static binary model only predicts it with 0.309 accuracy.

**Experiment Results** The Experiment 6 multimodal network combining multiple features from the facial images, landmarks, pain scores, sex, skin, and timeframe labels performs the best for overall pain classification. Compared to training on a single modality alone (Experiments 1, 2, 3, 7), Experiment 6 (Fusion 3) shows the best overall class accuracy of 0.657 shown in Table 5. Fusion 3 also shows the highest accuracy for the "Pain" level at 0.717. Experiment 6 (Fusion 3) led to 72.4% of test patients exceeding 50% accuracy per frame as noted in Figure 5b. However, it ties with Experiment

5 (Fusion2) and Experiment 2 (Random Forest PM) for 51.7% of test patients achieving over 75% accuracy per Figure 5a and Figure 4d. While the Random Forest pain model (Figure 5d) shows greater "No Pain" accuracy, using only the self-reported nine pain scores does not detect the original "Low" pain levels as well as the multimodal Fusion 3 model visualized in Figure 5c shown in blue bars.

Experiment 3 (Random Forest Pain) shows the highest "No Pain" accuracy scores at 0.706 per Table 5. Adding the nine self-reported pain scores appears to boost accuracy, compared to training only on faces and landmarks per Experiment 4 (Fusion 1, 0.513) in Table 5. This is likely due to high correlations between the nine reported pain scores. Analysis shows strong Pearson correlation values exceeding 0.89 among activity, mood, work, enjoyment, and relationship scores. Continued analysis as more patients enroll in the study is required to understand the effect of the nine pain scores across all patients. The facial landmarks perform the worst in Experiment 2 (Random Forest LM) with only 37.9% of test patients exceeding better than random at over 50% accuracy per Figure 4b. However, when adding landmarks to facial images in Experiment 4 (Fusion 1), several test patients completely fail to be detected (1, 2, 16, 13, 29, 3, 28, 25) per Figure 4d. This may be consistent with recent research by [34] who show that landmark detection declines when comparing different populations, such as older patients with dementia, to healthy adults.

Table 5: **Experiment Results by Pain Level Accuracy.** "LM" indicates facial landmarks.

| Experiment | 4-Class Model | Data | All Classes | None | Low | Moderate | Severe |
|---|---|---|---|---|---|---|---|
| Exp. 1 | ResNet50-4-static | Faces , only | 0.378 | 0.583 | 0.168 | 0.252 | 0.213 |
| | **2-Class Model** | | **All Classes** | **No Pain** | **Pain** | | |
| Exp. 1 | ResNet50-2-static | Faces, only | 0.568 | 0.513 | 0.612 | | |
| Exp. 2 | Random Forest LM | Landmarks, only | 0.373 | 0.479 | 0.287 | | |
| Exp. 3 | Random Forest Pain | Pain Scores, only | 0.650 | **0.706** | 0.602 | | |
| Exp. 4 | Fusion 1 | Faces + Landmarks | 0.513 | 0.304 | 0.683 | | |
| Exp. 5 | Fusion 2 | Faces + Landmarks + Pain Scores | 0.631 | 0.563 | 0.687 | | |
| Exp. 6 | Fusion 3 | Faces + Landmarks + Pain Scores | **0.657** | 0.582 | **0.717** | | |
| Exp. 7 | Static Audio | Audio, only | 0.456 | 0.645 | 0.303 | | |

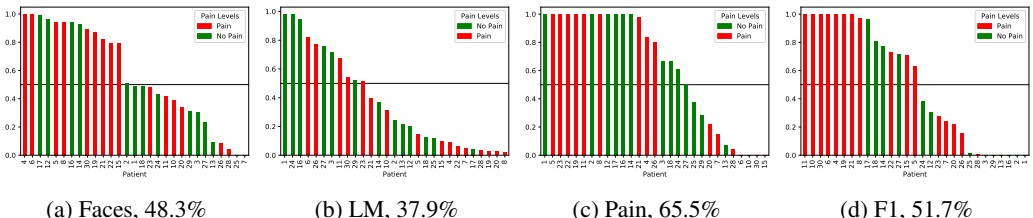

| (a) Faces, 48.3% | (b) LM, 37.9% | (c) Pain, 65.5% | (d) F1, 51.7% |

Figure 4: **Accuracy Scores per Test Patient by Model: Faces, Landmarks, Pain, and Fusion 1.** We show the resulting scores per test patient for the binary pain classifiers. Horizontal bar indicates 50% accuracy. Percentages in sub-captions indicates the number of patients exceeding 50% test accuracy. Notation: Faces=ResNet50-2-static; LM=Random Forest LM (landmarks); Pain=Random Forest Pain; F1=Fusion1. Best viewed in color and zoomed in.

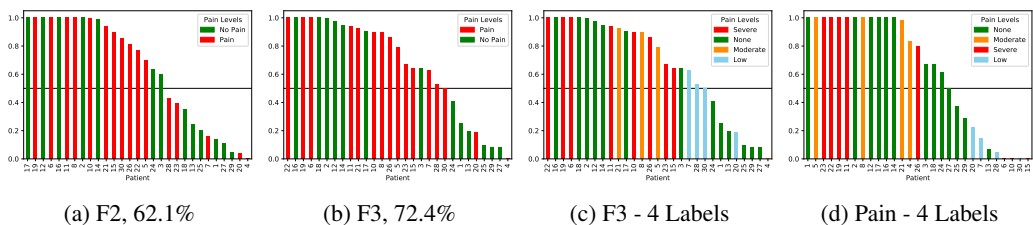

| (a) F2, 62.1% | (b) F3, 72.4% | (c) F3 - 4 Labels | (d) Pain - 4 Labels |

Figure 5: **Accuracy Scores per Test Patient by Model - Fusion 2, 3, and Pain, Visualized with 4 Original Labels.** We show the resulting scores per test patient for the binary pain classifiers. Horizontal bar indicates 50% accuracy. Percentages in sub-captions indicates the number of patients exceeding 50% test accuracy. Notation: Pain=Random Forest Pain; F1=Fusion1; F2=Fusion2; F3=Fusion3. Best viewed in color and zoomed in.

## 6 Discussion and Future Work

Due to a variety of state-of-the-art techniques, we sought to implement simple models to demonstrate baseline results using fairly minimal preprocessing, transformations, and architectures. The results of our models show the dataset's difficulty. For comparison, acute pain detection studies have shown accuracy scores up to 82.4% (hit rate) [26] using the UNBC-McMaster Shoulder Dataset and 95% for multimodal infant pain detection using a custom dataset by [31]. Chronic pain detection using psychological inventories have achieved 86.5% (cross-validated balanced accuracy) using a support vector machine [58].

**Limitations** The first limitation of the dataset is the low number of currently enrolled patients at only 29 patients and the imbalance across pain levels. However, we observe that two new patients enroll into the study every month. As the number of patients grow, we expect a more balanced distribution of pain levels, sex, skin type, and increased volume of data, consistent with the cohort design indicated in Table 2. However, medical datasets using active patient populations for major diseases such as cancer, are extremely scarce due to the time and review required for medical privacy and ethics. This differs greatly from current pain datasets that have recruited fairly healthy patients, who are not actively undergoing disease treatment. Due to the special sensitivity of the ISS study population, we believe that our current initial results offers important insights currently missing in the medical AI community.

Next, despite the patient instructions to complete the submission in a quiet, brightly lit room with a white wall or background, many videos submitted varied in quality and resolution. The following examples observed in the dataset present challenges to machine learning: 1) Patient sitting in front of a door with signage in the background showing letters and numbers; 2) Patient occasionally wears a mask in some videos (due to Covid-19); 3) Patient records video in area of intense sunshine and glare causing reflection from various surfaces; 4) Patient records in a dark, shady room, leading to grainy resolution and video quality; 5) Patient speaks very quietly or muffled, making it difficult to hear the patient narrative; 6) Missing data as is the case of Patient 0009 and absent self-reported nine pain scores from Patient 0015.

**Ethics** Publicly available acute pain datasets have lacked ethnic diversity. For example, the UNBC-McMaster Database [6] uses ethnicity as a demographic indicator where out of the original 129 patients (63 Male, 66 Female), a minimum of 13.2% (17 patients) consisted of non-Caucasian ethnicity (refer to Table 1 of [6]). It may be less given how studies using the UNBC-McMaster dataset have access to data from only 25 out of 129 patients [20–22, 24]. The BioVid and MIntPAIN datasets provide no information about ethnicity and race [8, 9]. EmoPAIN contains 22 patients (18 Caucasian, 3 African-American, 1 South-Asian) who are majority white [30]. As a result, we sought to increase the diversity of enrolled patients by using cohorts that include sex and skin type specifications. While the Fitzpatrick Skin Type scale was originally developed for dermatological use, it has recently been criticized for its conflation with race and ethnicity [59]. It has been found to overestimate the prevalence of Type IV skin classification in African Americans [60]. The visual grouping of patients into lighter tones (Skin Types I - III) or darker tones (Skin Types IV - VI) may be too restrictive and biased in terms of broadening our diversity of patients. As a result, the ISS dataset requires careful monitoring and a regular ethics review.

**Future Work** The second phase of the study will analyze more diverse modalities. First, we will extract text from the audio files and explore its utility towards multimodal pain models. Next, since patients were unable to conduct in-clinic visits, we were unable to gather thermal imagery captured from a thermal camera stationed at the clinic. Thermal imagery offers insights into physiological states that is unseen on visible images alone [61]. Our intent is to generate paired visible-thermal datasets as collected by the Iris, Eurecom, and Equinox datasets [62–64]. Lastly, we estimate that after enrolling 112 patients, the ISS dataset will contain an additional 1,456 videos, 543,733 frames, and 3.8 hours of content.

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
