# OpenReview forum: "Intelligent Sight and Sound: A Chronic Cancer Facial Pain Dataset"
_NeurIPS.cc/2021/Track/Datasets_and_Benchmarks/Round2 — NeurIPS 2021 Datasets and Benchmarks Track (Round 2)_

### Official Review · Reviewer_tYvH · 2021-09-17
**Review of "Intelligent Sight and Sound: A Chronic Cancer Pain Dataset"**

**Rating:** 7
**Confidence:** 3
**Correctness:** Yes, the dataset is constructed in a …
**Clarity:** The paper is written quite nicely.

**Strengths:**

1. The proposed dataset is very novel, which could benefit the research community.
2. The data collection part of the dataset is well presented with sufficient details.
3. The proposed dataset has diverse modalities, which makes it useful for the research community.



**Weaknesses:**

1. Presently, the dataset is collected from 29 patients.
2. The dataset is not public as the dataset is restricted in terms of access.
3. The dataset is restricted as it will be available on per-case basis.
4. Authors have excluded non-English speaking patients. Is there a reason for it?

**Additional Feedback:**

N/A

**Documentation:**

There is sufficient detail on the data collection and organization but lacks in terms of availability.

**Ethics:**

The collected dataset could possibly be biased to people from a specific region, or from English speaking population.

**Relation To Prior Work:**

The authors presented a discussion on how this work differs from previous contributions.

**Summary And Contributions:**

The authors collect and present a new dataset for detecting chronic pain levels for cancer patients in this paper. The dataset is proposed to be collected from 112 patients, but presently the dataset is collected from 29 patients. The dataset is collected from the patients who are getting treatment at NIH clinical center. The dataset contains patients' video, audio, and pain scores data. Dataset also has gender and skin tone information which makes it a diverse dataset.  The authors also trained baseline deep learning models for evaluation and reported the performance. The authors evaluated 2 class and 4 class classifications. The main contribution of this paper is the proposed dataset. Keeping novelty and diversity in mind, I think this would benefit the research community at large.

---

> ### Author Response · Authors · 2021-09-25
> **Response to Reviewer TYVH**
>
> We thank the reviewer for their valuable time, feedback, and insight. Please see answers provided below.
>
> 1 – Patient Size. Please see response to **Reviewer LSGP** Question 1 Patient Size.
>
> 2 & 3 – Please see response to **Reviewer 2873** Access to Code and Data Question 1.
>
> 4 - The prompt outlined in Section 3.2.2 Questions 10 and 11: Prompt and Narrative, are neutralizing baselines that need to be read aloud prior to making a video recording of the patient narrative. Since the narratives are in English, at this time, the NIH Clinical Team at the time of the protocol development and IRB, decided to exclude non-English speakers in order to ensure baseline prompts could be read. Further, we intend to transcribe the video for text and currently are planning to analyze English text.

---

> > ### Comment · Reviewer_tYvH · 2021-09-29
> > **Rating update**
> >
> > Thank you for your response. Based on the clarifications provided by the authors, I would update the rating to 7. Best of luck

---

> > > ### Author Response · Authors · 2021-09-29
> > > **Thank you for update**
> > >
> > > Thank you for reading all of our updated content and replies, we are very pleased it was able to satisfy your concerns. Please do not hesitate to let us know if you had any further questions.

---

### Official Review · Reviewer_2873 · 2021-09-18
**Interesting and comprehensive multimodal dataset**

**Rating:** 6
**Confidence:** 4
**Clarity:** The language is clear and easy to und…

**Strengths:**

* The paper is well structured and written.
* Much thought is given to data fairness and ethical considerations.
* The dataset includes a wide array of modalities, providing value to multiple areas of research within machine learning.
* Demographic data is reported and well distributed over age, sex, and skin tone.
* Authors offer a thorough description of the annotation protocol and platform used by the patients to collect data.


**Weaknesses:**

* The code used in this paper is not made public, thus making it impossible to reproduce the baselines even if one obtains permission to use the data.
* Slightly more information about the dataset should be offered for it to be useful in a practical setting. In particular, the distribution of source devices and the patient level variance in pain levels would be helpful for future work.
* No info about how hyper parameters were chosen is given, and no replications of the experiments were done such that readers can understand the expected variance within the methods.
* If I understand the text correctly, for experiments 3, 5, and 6, the authors use the self-reported pain levels as metadata to help predict the pain score. However, as the pain score is derived from the self-reported pain levels, this metadata element can be seen as severe data leakage, i.e. the model may learn the mapping between the self-reported pain level and the pain/no-pain target label instead of using the rest of the data to predict the pain/no-pain target label.


**Additional Feedback:**

While the dataset seems to be a valuable medical resource, the experiments section feels lacking to me, primarily due to the use of the self-reported pain levels to predict the pain value.

**Correctness:**

Data is collected using a thorough protocol. Experiments are comprehensive, but cannot be reproduced. If my understanding is correct, some experiments train on a meta-data variable that is used to define the target label.

**Documentation:**

While the annotation protocol is well documented, no code is made publicly available.

**Ethics:**

All concerns were addressed.

**Relation To Prior Work:**

The paper is well placed within the current state of the art.

**Summary And Contributions:**

The authors propose a new multimodal facial cancer pain detection dataset, where the pain levels are self-assessed by the patients. Modalities include video, associated metadata, and audio files. All data is collected by the 29 patients included in this dataset, that record themselves several times a week over several weeks. This data is then used to predict the pain level using several different methods, ranging from random forests to convolutional neural networks.

---

> ### Author Response · Authors · 2021-09-25
> **Response to Reviewer 2873**
>
> We thank the reviewer for their generous time and insight.
>
> 1 – Access to Code & Data. When researchers reach out to request access to the data, they will also receive the code for the benchmark models (we are working to get the code online w/o request, but must ensure no accidental PHI). We also provide in the Appendix "Section F.2 Model Architectures" the diagrams for each network.  In "Consent to Participate in an NIH Clinical Research Study" provided at the end of the Supplementary Materials, it states that the data is subject to strict medical privacy requirements per the Federal Privacy Act. Legal protections are promised to patients enrolled in the study which cannot be violated. Further, NIH policies require data to be stored in an internal NIH database. However, we will provide a way to contact the Principal Investigator, Dr. James Gulley, to access the ISS dataset and code, upon publication of the paper which may be granted on a per-case basis. Given the ethical discussions being had in the AI community to date, we are ensuring a balance between ethical privacy protections and advancement for the scientific community.
>
> 2 & 3 – Source Devices and Hyperparams. Thank you for this insightful request. We have data about the source devices and will include it in the Appendix (i.e. distribution of Android, iOS, clinical-only). Regarding patient pain level variances, **please see response to Reviewer JKYC Question 3 Pain Variance.** Regarding hyperparameters, we selected the default values for the Random Forest Classifier (100 estimators, gini criterion, etc.). For the neural network experiments, we empirically tested the selection of hyperparameters listed in Section 4 Baselines ``Training Details" which are consistent across all experiments. The batch size of 16, which was selected empirically to yield the best cross validation accuracies, after running several experiments varying batch size from 4, 8, 16, 32, and 64. We ran experiments using both Stochastic Gradient Descent with 1e-4 learning rate and 0.9 momentum, in addition to Adam optimizer with 1e-4 learning rate and found that there were few differences. Adam optimizer has been used in additional facial pain classification studies to include Bellantonio and Rodriguez. Overall, the time it takes to train these seven experiments was computationally expensive due to 10-fold cross-validation. With a low sample size and high cost, we found that additional hyperparameter tuning was inadvisable.
>
> 4 - Pain Scores and Leakage. We apologize for the confusion and will clarify in our paper the difference between the nine self-reported pain scores and the **pain target**.  Experiments 3, 5, and 6 use nine self-reported pain scores from the BPI (Section 3.2.1, Questions 1 - 9: Self-Reported Pain Scores) as inputs to the model. The pain target is **not** one of these nine self-reported scores. For example, in Table 2, the four "Self-Rep. Pain" levels 0, 1-3, 4-6, and 7-10, are assigned at the start of the study for each patient and does not change through their 3-mos. period. We will change this name in Table 2 as ''Pain Target" to prevent confusion. But, the nine self-reported pain scores do change. These scores change at every video submission and are distinct and unrelated to the target pain level. There is no formula that relates the nine self-reported scores among themselves or to the pain score target. These questions focus on emotional and activity pain asking the patients to rate their levels of pain by how it interferes with issues like mood, walking ability, housework, and sleep.  The use of ``worst possible pain" may be the confusing term, where on lines 87 - 88 we mention that the patient reports their worst pain level. Lines 87 - 88 are actually referring to lines 102 - 103, the worst pain the patient has experienced in the past 30 days. Please see response to Reviewer JKYC Motivation Question 2, as to why including these emotional pain scores are valuable.
>
> Add. Feedback - We contend that the use of self-reported pain is an extreme benefit of the data itself. As mentioned in ``Section 3.3.2 No External Labels", the current works on pain use measures of pain that are not clinically relevant or biased toward an observer's impression of a patient's pain. Yet, subjective pain is the clinically relevant target that medical providers are trying to manage,  and can only be obtained from patient response. We highlight a related example pointed out by Kate Crawford ( \url{https://www.nature.com/articles/d41586-021-00868-5}) in the use of observed-based assumptions leading to the applications of methods that were ultimately not valid, because the external labeling encodes viewer bias, when the underlying goal is more nuanced. In our case, the goal of knowing the level of pain can only be retrieved from the patients themselves, and we hope evidence can be built for an ML approach to still be effective with this intrinsically subjective yet universal goal.

---

> > ### Comment · Reviewer_2873 · 2021-09-29
> > **Rating update**
> >
> > Thank you for your thorough response. Based on the clarification offered in your reply, as well as the updates to the paper, I will change my rating to a 6.
> >
> > I am looking forward to seeing the full version of this dataset, once the study is complete.

---

> > > ### Author Response · Authors · 2021-09-29
> > > **Thank you for reading revisions & reply**
> > >
> > > We thank the reviewer for reading our rebuttals and revised paper, especially for the raised score! While COVID has delayed data collection, we are continually enrolling and collecting new patients, and confident early access to the data and its regular updates over time will be valuable to those in the CV community looking to tackle challenges in multi-modal sample efficiency and work toward a positive impact on society.

---

### Official Review · Reviewer_LSGP · 2021-09-19
**The paper presents a multimodal dataset for assessing pain in patients suffering from cancer**

**Rating:** 6
**Confidence:** 3
**Correctness:** The papers seem correct.
**Clarity:** The paper is written vey clear.

**Strengths:**

The paper presents very detailed the protocol for collecting data in the multimodal setting: audio, video and pain scores. Apparently, there are no datasets for the task of pain assessment for patients suffering from cancer. The  experimental section seems well conducted and show evidence for the usefulness of the dataset. In terms of data volume, this data compares to [7], but for a different task.

**Weaknesses:**

My main concerned is the small number of patients currently included in the study, only 29. As Tabel 2 shows there are currently combinations of score pains, skin types and sex not covered by the existing patients. As far as I know, usually clinic studies consists in a moderate number of patients, far from large-scale.

**Additional Feedback:**

See weaknesses.

**Documentation:**

Seem fine.

**Ethics:**

There seems to be no problems.

**Relation To Prior Work:**

Seems well covered.

**Summary And Contributions:**

This paper introduces a dataset for assessing the level of pain for patients suffering from cancer . The datasets consist of self reported pain scores, video and audio data gathered from 29 patients and comprise 509 videos with 190k video frames. The proposed baselines show that combining faces, landmarks extracted from faces and pain scores the task of assessing pain cand be tackled with moderate success.

---

> ### Author Response · Authors · 2021-09-25
> **Response to Reviewer LSGP**
>
> We are grateful for the reviewer's feedback and generous time. Please find our responses to concerns about the patient size.
>
> 1 – **Patient Size**. We appreciate the reviewer's comments and understand the concerns about patient size. We point out in the "Section 6: Discussion and Future Work" that we are currently limited in this paper by 29 patients. However, as we indicate in "Future Work", we onboard new patients weekly. Currently we have onboarded an additional four patients. But despite the growing number of enrolled patients, we emphasize that it was extremely difficult to have even recruited and carefully designed the data for these 29 patients, particularly in an attempt to account for gender and ethnic diversity. Medical datasets using active patient populations for major diseases such as cancer, are extremely scarce, where the time it takes to carefully review data for medical privacy and ethics takes a considerable amount of time.
>
> This differs greatly from current pain classification studies that have recruited fairly healthy patients, who are not actively undergoing disease treatment. Due to the special sensitivity of this population, cancer patients who are actively undergoing treatment, we believe that our current initial results offers important insights currently missing in the medical AI community.  In addition, for the reviewer's consideration, the UNBC-McMaster dataset is currently SOTA for pain detection. It consists of 25 patients (four less than our current paper), with almost one-quarter the number of frames as our dataset, which implies less data for fewer patients, when compared to the ISS dataset.
>
> Ultimately, we do not feel a large dataset (e.g., ImageNet scale) will be realistic for carefully crafted and clinically relevant data. Getting transfer learning, or other sample efficiency approaches, to work on data such as our own is an important research problem that our data enables study of from multiple modalities. The AI/ML community has made significant progress in computer vision thanks to ImageNet and even larger datasets, in part because the simplicity of the datasets themselves. The ambiguity is low, labeling them requires limited training or special skill, and the relative risk to image content is relatively low (though not trivial). In our effort to build this dataset for the community, the only of its kind as far as we are aware, we will be forced to rely on advancements in sample efficiency, transfer learning, and Bayesian analysis, to make robust conclusions and clinically relevant progress.

---

### Official Review · Reviewer_JKyc · 2021-09-20
**introduces a pain classification dataset, but can benefit from improvements on the experiments and dataset curation**

**Rating:** 5
**Confidence:** 4

**Strengths:**

The work introduces a dataset for chronic cancer related pain classification, which the authors claim is the first of its kind. It is very much appreciated that the authors use the main text and appendix to point to current limitations of pain prediction as related to skin color and ethnicity and it should be lauded that the authors are taking these factors into consideration for their dataset

**Weaknesses:**

1.	The current size of the dataset makes it difficult to make statistically powerful claims about takeaways from the data, especially because the dataset is a classification dataset. Even with cross validation, a total of 29 observations may be too small for any statically powered comparison. Given that the authors plan on expanding this dataset in the future, I would recommend that the authors collect the data and run this analysis with more participants.
2.	While the dataset is novel, it may be unclear to readers why pain classification is an important problem. There have been previous studies that have indicated biases in pain characterization among medical professionals, but given that this dataset tackles self-reported pain, it would be useful for the authors to describe the need to automated pain classification in this setting.
3.	Especially given that this dataset operates in a classification setting, it would be useful to
4.	The work may benefit considerably by considering longitudinal pain tracking for subjects. Given that subjects are recorded multiple times over a long time period, such an analysis is feasible and could help address subject-specific analysis questions – eg. a) are auditory/facial features correlated with subject-specific changes in pain.
5.	The reporting of metrics is also a bit confusing. The authors mention that “pain level accuracy scores are the average of patients’ accuracy scores representing each pain level”. Is there a particular reason this is done as opposed to reporting the classification (0/1-binary, 0/1/2/3 – multi-class) score for a subject?
6.	Certain methods around the computation of pain scores may need to be clarified. It is not clear how the 4 pain levels if these levels are a result of the questionnaire responses, and if so, how they are determined from the questionnaire.
7.	Experiments 4-6 use pain score as an input into the model. this may very likely be conflated with the 4-class and 2-class pain score outputs that the model needs to predict. This would give an undue advantage to these models in performance, which may contribute to the improved performance. In practice, it may be prudent to ablate this parameter to truly understand it’s impact on performance. If it is known a priori that the output labels are likely a function of the input pain scores (which seems likely), the pain scores should not be used as an input as this could lead to a circular problem.
8.	The work can also improve with clarity in terms of how the test results were computed. For example, for experiment 1, is each frame for a patient classified, and the resulting probability averaged per subject to get a classification label?
9.	Please also indicate if the subjects are participants or patients


**Additional Feedback:**

N/A

**Clarity:**

The paper is mostly free of grammatical errors but could benefit from a more detailed description of the experimental setup.

**Correctness:**

There are some concerns I have with the correctness of the experimental setup, especially Experiments 4-6 as detailed above.

**Documentation:**

There is no documentation on how to access the dataset at this time or how to reproduce benchmarks as no code is provided.

**Ethics:**

The work mentions ethical considerations in the curation and release of their dataset. It is still unclear as to how this will impact the usability of and access to the dataset.

**Relation To Prior Work:**

The work gives a reasonable summary of the related work

**Summary And Contributions:**

This paper introduces a dataset for chronic cancer-related pain classification. The dataset consists of questionnaires and smartphone videos collected from 29 patients over a period of 8 months. The work also evaluates baseline models for pain classification using both unimodal and multimodal features extracted from image frames and audio in smartphone videos.

While the work makes a concerted effort in introducing this new dataset, it could benefit substantially from:
1.	A more detailed and sound analysis of the dataset and how users can leverage the data
2.	Waiting for more data to be collected for more statistical power for the dataset and baseline methods

---

> ### Author Response · Authors · 2021-09-25
> **Response to Reviewer JKYC**
>
> We appreciate Reviewer JKYC's generous time and valuable comments. Please find our response below.
>
> 1 – **Patient Size**. Please see response to **Reviewer LGSP.**
>
> 2 – **Motivation and Self-Reported Pain**. We briefly touch on this in "Section 3.3.2 No External Labels" and will revise to clarify the reasons promoting the use of self-reported pain labels as opposed to externally encoded ones. Our motivation lies in the lack of chronic cancer pain classification, which is an important problem for physicians who regularly treat cancer patients who may lack healthcare resources, be non-communicative, traumatized, and emotionally restrained due to adverse mental and physical health issues. Managing cancer patient pain is a part of a larger medical calculus in delivering precise services, treatment, and medicines as outlined in the 1994 AHCPR Cancer Pain Guidelines (Schug). Sub-optimal pain management can block patient recovery and improvement, making the already difficult cancer experience, worse, for both patient and family (Sun and Cleary). Manual clinical assessment requires accounting for a landscape of complex emotions and beliefs that clinicians must regularly take into account when assessing cancer patient pain - physical, psychological, social, and spiritual elements combined with severe distress for future outlook. For example, patients undergoing chemotherapy are more likely to believe that ``good patients" do not complain about pain which they believe can be distracting to clinicians and be non-communicative. Very few patients are actually screened for pain at each clinical visit and Cleary et al. indicates that pain is also underreported in different patient populations. Due to the variety of complex conditions affecting cancer pain, experts recommend repeated, regular pain assessment, which can be difficult and impractical for manual assessment by clinicians.  From a methodological perspective, current pain classification has been researched by large, for acute, musculoskeletal pain such as painful shoulder movements or lower back pain. As a result, a natural reaction in pain classification studies may be to invest a great deal of resources towards external FACS labeling. Here, onlookers gauge the level of pain which can be assessed more objectively when there are extreme facial expressions for pain. However, limited works show that this is successful for predicting chronic health pain, where facial expressions are typically placid, subdued, and motionless. It is important to point out that we explore self-reported pain across not only faces (image modality), but different modalities such as additional emotional and activity pain scores, landmarks, and audio, in an attempt to collect as much information about the cancer patient experience to classify pain.
>
> 3 -  **Pain Variance**. The self reported level of pain is determined at study onset and is chronic, not varying during the 3 mos. of patient enrollment. This means a study of pain fluctuations over time is not possible with our data. We will make this clearer in revision, and further explain how this was conducted following standard medical research protocols in response to your 5th question. However, we appreciate your recommendation which would be plausible for the nine other self-reported pain scores. We will confer with clinicians and the current literature to see if this would be a clinically relevant outcome to predict.
>
> 5 – **Pain Score Methods & Clarification of Pain Scores and Leakage**. The four pain levels are not a part of the questionnaire and there is no formula to calculate them. These pain levels are the pain target for classification, assigned at enrollment, and are distinct from the other nine self-reported pain scores.  In detail, please see response to Reviewer 2873
>
> 4 & 7 - Regarding **calculation of accuracy**, we will clarify this in the paper. In detail, we calculate results in the following manner:
> 1.	We performed 10-fold cross-validation (App Section F.1. Table 10).
> 2.	Using the tenth checkpoint, we evaluate each test patient individually. We only evaluate test patients using the correct assigned split, otherwise this would result in target leakage. For example, referring to Table 10, Test Patients 0002, 0029, and 0021 are only evaluated using the trained model from Split 1, not Split 2 which would have included these three patients in its training set.
> 3.	We evaluate each test patient using a batch size of 1.
> 4.	We then calculate accuracy for the test patient in question as simply accuracy_score(y_true, y_pred) where y_true are true pain target labels and y_pred are pain target predicted labels.
>
> 8 - We interchange these terms by mistake and will use the term ``patients" moving forward. All patients are indeed patients who are actively undergoing cancer treatment.
>
> 9 – **Access to Code and Data**. Please see response to **Reviewer 2873**.

---

> > ### Author Response · Authors · 2021-09-26
> > **References**
> >
> > Please find the following references for citations in the previous reply:
> >
> > Schug, Stephan A., Detlev Zech, and Ulrike Dörr. "Cancer pain management according to WHO analgesic guidelines." Journal of pain and symptom management 5.1 (1990): 27-32.
> >
> > Sun, Virginia Chih-Yi, et al. "Overcoming barriers to cancer pain management: an institutional change model." Journal of pain and symptom management 34.4 (2007): 359-369.
> >
> > Cleary, James F. "Cancer pain management." Cancer Control 7.2 (2000): 120-131.

---

> > ### Comment · Reviewer_JKyc · 2021-09-30
> > **Updated Rating**
> >
> > Thank you for your clarification and comments. I have updated my rating as a result of the resolution of most of the comments
> >
> > ### Self-Reported Pain Scores
> > As the authors mentioned, including details about issues with self-reported pain (eg. underreporting) would be very helpful for clarifying the utility of self-reported pain classification.
> >
> > The authors mention that experts recommend repeated and regular pain assessment is critical as self-reported pain values may fluctuate, but also state that the pain values in this study do not fluctuate because the pain is chronic. However, as noted by the references provided by the authors, pain values do fluctuate even in chronic pain, though the overall bias/average pain value reported may be stable.
> >
> > In such a scenario, it would be useful for the authors to explain the design choice of acquiring a single self-reported pain value for 3 months for each patient. While this may be part of the submitted protocol for acquiring data, it raises some potential concerns about the accuracy of the correlation and labels between facial/audio/activity features and one-time self-reported pain values
> >
> > 1. **Longitudinal Correlation between Facial Features and Self-Reported Pain**: As mentioned by the authors and prior literature they cite, self-reported pain values can be variable among patients. Thus features for a single patient may change throughout the data acquisition process. Thus, features acquired at timepoint of 1 month may not be indicative of their pain level at the start of the study.
> >
> > 2. **Label Accuracy**: If pain values are reported among particular populations of data, similar facial expressions can result in vastly different pain scores which, in a field of 29 patients, is difficult to adequately characterize. If each patient feels and reports pain based on a subjective interpretation of the scale and what they are comfortable reporting, the accuracy of labels in a small cohort is also difficult to quantify. Of course, there is no real way to establish a standard scale for pain or evaluate accuracy of the labels. As such, it would be useful for the authors to mention this as a limitation of the dataset and provide a discussion for how this could be addressed for future versions of the dataset.
> >
> > One advantage of longitudinal pain score collection is that deviations from a patient's pain baseline, rather than singular predictions, can be quantified. As such some of the following analysis can be enabled:
> >
> > 1. How variable is the patient's pain?
> > 2. Does the patient have similar facial features / predicted values, but different self-reported pain scores?
> >
> > While I understand that the motivation behind this and other pain classifiers, it is imperative to note that pain is not an objective value, as reported by the references the authors mention. Even in cases of chronic pain, pain levels may fluctuate and thus impact the observable qualities of a patient.
> >
> > ### Code/Data Availability
> > While it is perfectly understandable that the data cannot be released publicly, it would be helpful to have clarity around what this maintenance plan and code would look like, what the license for data would be, and how new versions of the data would be distributed. From a reviewer's point of view, it is difficult to vet the quality and structure of the data without having some way to inspect the dataset on review. Because this is something that does not appear to be feasible at this time, I appreciate the authors' remarks and hope to see a version of this dataset in the near future. it would be useful for the authors to provide some specifics into the access protocol in the text (e.g. licensing).

---

> > > ### Author Response · Authors · 2021-09-30
> > > **Further Clarification.**
> > >
> > > We appreciate the raised score and thank the reviewer for taking the time to read our revision and rebuttals. We hope the further clarification below will fully satisfy your remaining concerns.
> > >
> > > We clarify for the reviewer, and will further clarify in the paper, that the primary "pain target" level represents the worst level of chronic pain reported by the patient in the past 30 days, which lasts over the 3-month period (i.e., at enrolment they report worst pain in past 30 days, and prior work has shown that score is stable for the 3-month window of the study). Per our previous citations, this is standard practice to measure the past 30 days of pain and track on a window of 3-months, and is guided by clinical oncologists based on current pain research. Some references showing that pain is standard to be reported in the past 30 days is provided at the end of this response. The "pain target" increases the window of time that the patient report is relevant and is slow to change beyond the 3-month time frame. The other nine pain scores that we collect are much more frequent and capture variable scores of pain such as its influence on emotion and activity, which change. We have further anecdotal evidence of this being confirmed in clinical follow up data, but can not report it as it falls outside of what the IRB approved and patients agreed to. However, we believe the below references are stronger evidence that attest to the validity of this protocol.
> > >
> > > We will reinforce in the paper that overall, pain has no true objective scale or determination, and can change over time. However, our current protocol is led by domain experts (clinical oncologists) designed to follow the best current scientific understanding of chronic pain. We believe that the protocol they designed would lead to analysis of relevant chronic pain outcomes, currently not studied in AI today. We also believe that, as the reviewer pointed out, a longitudinal study of the nine additional pain scores (emotion, activity, etc.) could be analyzed due to its variability. However, the "pain target", again, is fixed to a 3-month window.  To perform a longitudinal study of the "pain target" would require patients to be enrolled longer in the study (i.e. such as an additional 3 months) which would not be feasible due to risks of patient retention and deteriorating patient health.
> > >
> > > On label accuracy, we will further strengthen the point that prior datasets use external estimates of pain rated by observers. Such labels are not clinically relevant and often biased, and intrinsically lacks clinical relevance for a variety of reasons (unobtainable, biased, etc). Patient self reports of pain who are known to have a condition that causes pain (i.e., cancer) with carefully designed survey language is not perfect, but the most accurate and clinically relevant label that is obtainable and what we provide.
> > >
> > > Data availability will require the requesting participants to be apart of a academic or industrial institution, require evidence of completed training on Human Subeject Research, HIPPA, and all relevant legal trainings for laws governing the sensitive nature of the subjects. Access will not allow redistribution of the data, it must be obtained via the NIH per law, agreement with study participants, and contractual requirements. There will be a fast-track for data access approval for intramural researchers with already existing NIH work or NIH association to enable faster access to a wider breadth of academic and industry researchers already in this or adjacent spaces. These restrictions are again necessary under U.S. law, IRB approvals for this research, and to protected the highly sensitive nature of cancer patient data.
> > >
> > > Refs:
> > > Daut, Randall L., and Charles S. Cleeland. "The prevalence and severity of pain in cancer." Cancer 50.9 (1982): 1913-1918.
> > > Baker, K., et al. "Relation of synovitis to knee pain using contrast-enhanced MRIs." Annals of the rheumatic diseases 69.10 (2010): 1779-1783.
> > > Bolen, Julie, et al. "Peer reviewed: differences in the prevalence and impact of arthritis among racial/ethnic groups in the United States, National Health Interview Survey, 2002, 2003, and 2006." Preventing chronic disease 7.3 (2010).

---

### Author Response · Authors · 2021-09-29
**Updated Manuscript**

We would like to let the reviewers know that we have updated the manuscript to incorporate your valuable feedback. We hope the update and our rebuttals have satisfied your concerns, and we hope the reviewers will agree with the value of the dataset. Developing datasets with clinical relevance and accounting for bias is an extraordinarily labor intensive and interdisciplinary work, and also requires extensive care and legal requirements to protected the PII nature of the data. We believe this corpus will be of value for many computer vision researchers wishing to tackle important health applications, drive and motivate the need for sample efficient and better transfer learning, and serve as a valuable case study in design and data collection challenges in sensitive medical contexts.

---

### Decision · Program_Chairs · 2021-10-09

**Decision:**

Accept

**Comment:**

The problem and data is of extremely relevance. However all reviewers agree on some major drawbacks: limited size of the data, data sharing limitations, and some missing information. After discussion with the authors several critiques have been clarified. Overall the paper achieves the minimum score to be accepted for publication at the NeurIPS data track.